# Predictors of the post-COVID condition following mild SARS-CoV-2 infection

B-A. Reme [ORCID] [1,2] ✉, J. Gjesvik[1,3] & K. Magnusson [ORCID] [1,4]

Whereas the nature of the post-COVID condition following mild acute COVID-19 is increasingly well described in the literature, knowledge of its risk factors, and whether it can be predicted, remains limited. This study, conducted in Norway, uses individual-level register data from 214,667 SARS-CoV-2 infected individuals covering a range of demographic, socioeconomic factors, as well as cause-specific healthcare utilization in the years prior to infection to assess the risk of post-COVID complaints ≥3 months after testing positive. We find that the risk of post-COVID was higher among individuals who prior to infection had been diagnosed with psychological (OR = 2.12, 95% CI 1.84–2.44), respiratory (OR = 2.03, 95% CI 1.78–2.32), or general and unspecified health problems (OR = 1.78, 95% CI 1.52–2.09). To assess the predictability of post-COVID after mild initial disease, we use machine learning methods and find that pre-infection characteristics, combined with information on the SARS-CoV-2 virus type and vaccine status, to a considerable extent (AUC = 0.79, 95% CI 0.75–0.81) could predict the occurrence of post-COVID complaints in our sample.

An increased number of medical complaints following a mild SARS-CoV2 infection, frequently referred to as post-COVID or long COVID, has been reported in several studies[1–5]. The reported prevalence of post-COVID health complaints varies considerably, depending on its definition in terms of duration, the sample used (e.g., hospitalized or non-hospitalized), and method of measurement (e.g., self-report or register data), in the range of low single digits to well above 50 percent[6,7]. According to the WHO's Delphi consensus, a post-COVID condition occurs if covid-like complaints are present 3 months after infection, lasts for at least 2 months, and cannot be explained by any other diagnosis[8]. Although the types of complaints vary, the most common phenotypes include shortness of breath, loss of taste or smell, brain fog and fatigue[9–12].

While a large literature examines the prevalence of post-acute complaints[13–15], less is known about the incidence of doctor-diagnosed post-COVID condition and how well it can be predicted for individuals based on the presence of individual risk factors or risk factors in combination[12,16,17]. For example, regarding demographic characteristics, a higher persistence of symptoms was found among women and a lower persistence among younger individuals[4,6,12,18–20]. The evidence regarding the role of socioeconomic characteristics, however, is mixed[4,6]. For example, in a retrospective matched cohort study of 486,149 individuals in the UK, there was an increased risk of 21% (aHR 1.21, 95% CI 1.10–1.34) among ethnic minorities, and a 11% higher risk among those most socioeconomically deprived (aHR 1.11, 95% CI 1.07–1.16)[4]. However, a study combining 10 UK longitudinal studies and electronic health records from primary care, find evidence suggesting a higher prevalence of post-COVID complaints among white and highly educated[6]. Other studies find no socioeconomic gradient in post-COVID health complaints[21]. Emerging evidence also suggests that pre-existing poor health increases the risk, with higher risks among individuals with asthma, obesity, and psychological problems[4,6,18–20]. To what extent such pre-existing health, socioeconomic, and sociodemographic characteristics when studied in combination can predict doctor-diagnosed post-COVID condition in accordance with the WHO definition is unclear.

Obtaining estimates of the incidence of the post-COVID condition, as well as being able to accurately predict the probability of doctor-diagnosed post-COVID condition, may also be important to prevent long-term illness, sick leave, and disability. For example,

¹Cluster for Health Services Research, Norwegian Institute of Public Health, Oslo, Norway. ²Department of Health Management and Health Economics, University of Oslo, Oslo, Norway. ³Section for Breast Cancer Screening, Cancer Registry of Norway, Oslo, Norway. ⁴Clinical Epidemiology Unit, Orthopedics, Department of Clinical Sciences Lund, Lund University, Lund, Sweden. ✉e-mail: Bjorn-Atle.Reme@fhi.no

knowing upfront that an individual with COVID-19 is at heightened risk of post-COVID condition may aid clinicians to take early action to limit long-term consequences, e.g., through early referral to rehabilitation. Thus, we had the following two aims: (1) To assess the incidence and risk of doctor-diagnosed post-COVID condition for up to six months after positive test by characteristics of pre-existing poor health, socioeconomic and sociodemographic factors, and (2) to develop prediction models of doctor-diagnosed post-COVID condition using machine learning which combines detailed individual-level demographic, socioeconomic and health care utilization data.

## Results

### Descriptive statistics

Of 238,001 eligible participants, we included 214,667 individuals. Thus, 23,334 individuals were excluded due to missing on education or income, being hospitalized, or infected twice within 180 days (Supplementary Fig. 1). The mean (SD) age was 44.6 (9.8) years and 50% were women. In total, 0.42% (N = 908) were classified as having a post-COVID condition (main outcome). Among these participants, 206 (21%) were classified as experiencing post-COVID respiratory complaints, while 584 (60%) were classified as experiencing post-COVID fatigue (cf. Supplementary Table 1). Of the 206 individuals with post-COVID respiratory complaints, 191 (93%) were new onsets compared to the period between 2017 and 2019 (Supplementary Table 1). Similarly, out of the 584 individuals with fatigue, 444 (76%) had new onsets of fatigue complaints. Hence, for most participants these complaints were more likely to be due to the infection and not already preexisting conditions.

Table 1 presents descriptive statistics for the variables used in the main analysis, both for the whole sample and for each outcome group. Supplementary Fig. 2 shows that the majority of the included individuals had their pre-infection healthcare utilization measured approximately two years ago, and a smaller part had it measured zero to one year ago. Descriptive characteristics by pandemic period (based on virus dominance) showed that the pre-pandemic healthcare utilization was balanced across groups, i.e., not dependent on the time interval passing between the date of SARS-CoV-2 infection and registration of previous complaints/healthcare use (Supplementary Table 2).

### Demographic, socioeconomic, and COVID-related risk factors

The strongest bivariate associations were found for female (OR = 2.17, 95% CI 1.89–2.50) and infection by with the Wuhan virus type (OR = 4.00, 95% CI 3.48–4.6) (Fig. 1). When adjusting for prior health care utilization (the multivariate models), the OR was reduced to 1.93 (95% CI 1.67–2.23) for women. When adjusting for prior health care utilization and differences in vaccination across the virus types, the OR for the Wuhan-virus was reduced to 2.27 (95% CI 1.90–2.71). Vaccination was strongly (negatively) associated with the post-COVID condition in the bivariate model (OR = 0.33, 95% CI 0.29–0.38), however, the association was not present in the multivariate model, which included virus types (aOR = 1.01, 95% CI 0.80–1.28).

### Prior healthcare utilization risk factors

The strongest bivariate associations were found for Psychological (OR = 2.12, 95% CI 1.84–2.44), Respiratory (OR = 2.03, 95% CI 1.78–2.32) and General and unspecified (OR = 1.78, 95% CI 1.52–2.09) health problems. These findings imply that individuals who prior to the pandemic had a psychological diagnosis were approximately twice as likely to be classified with the post-COVID condition, compared to infected individuals without such prior diagnoses. When adjusting for demographic and socioeconomic factors, the strongest associations were found for Respiratory (aOR = 1.93, 95% CI 1.69–2.12), Psychological (aOR=1.81, 95% CI 1.57–2.09), and Digestive (aOR = 1.69, 95% CI 1.44–1.98) health problems.

**Table 1 | Descriptive statistics – number of observations and incidence**

| | All (N) | Not Post-COVID (N) | Post-COVID (N) | Incidence of post-COVID condition within the respective predictor % (95% confidence interval) |
|---|---|---|---|---|
| All (baseline incidence risk) | 214,667 | 213,759 | 908 | 0.42% [0.4%–0.45%] |
| Male | 107,123 | 106,837 | 286 | 0.27% [0.24%–0.3%] |
| Female | 107,544 | 106,922 | 622 | 0.58% [0.53%–0.62%] |
| Non-Immigrant | 147,072 | 146,358 | 714 | 0.49% [0.45%–0.52%] |
| Immigrant | 67,595 | 67,401 | 194 | 0.29% [0.25%–0.33%] |
| Age group [30,40] | 85,473 | 85,151 | 322 | 0.38% [0.34%–0.42%] |
| Age group ⟨40,50] | 70,751 | 70,425 | 326 | 0.46% [0.41%–0.51%] |
| Age group ⟨50,60] | 41,426 | 41,223 | 203 | 0.49% [0.42%–0.56%] |
| Age group ⟨60,70] | 17,017 | 16,960 | 57 | 0.33% [0.25%–0.42%] |
| Education: Primary | 42,750 | 42,610 | 140 | 0.33% [0.27%–0.38%] |
| Education: Secondary | 73,980 | 73,683 | 297 | 0.4% [0.36%–0.45%] |
| Education: Low Uni | 65,045 | 64,684 | 361 | 0.56% [0.5%–0.61%] |
| Education: High Uni | 32,892 | 32,782 | 110 | 0.33% [0.27%–0.4%] |
| Income percentile ⟨0,20] | 41,944 | 41,838 | 106 | 0.25% [0.2%–0.3%] |
| Income percentile ⟨20,40] | 41,090 | 40,937 | 153 | 0.37% [0.31%–0.43%] |
| Income percentile ⟨40,60] | 43,654 | 43,414 | 240 | 0.55% [0.48%–0.62%] |
| Income percentile ⟨60,80] | 43,711 | 43,478 | 233 | 0.53% [0.46%–0.6%] |
| Income percentile ⟨80,100] | 44,268 | 44,092 | 176 | 0.4% [0.34%–0.46%] |
| **Virus and vaccine** | | | | |
| Vaccine[a] | 150,281 | 149,882 | 399 | 0.27% [0.24%–0.29%] |
| Wuhan virus | 22,945 | 22,653 | 292 | 1.27% [1.13%–1.42%] |
| Alpha virus | 22,508 | 22,347 | 161 | 0.72% [0.61%–0.83%] |
| Delta virus | 78,518 | 78,171 | 347 | 0.44% [0.4%–0.49%] |
| Omicron virus | 90,696 | 90,588 | 108 | 0.12% [0.1%–0.14%] |
| **Healthcare utilization 2017–2019** | | | | |
| General and Unspecified (A) | 28,649 | 28,454 | 195 | 0.68% [0.59%–0.78%] |
| Blood, Blood Forming Organs and Immune Mechanism (B) | 2988 | 2974 | 14 | 0.47% [0.22%–0.71%] |
| Digestive (D) | 30,536 | 30,337 | 199 | 0.65% [0.56%–0.74%] |
| Eye (F) | 8326 | 8273 | 53 | 0.64% [0.47%–0.81%] |
| Ear (H) | 7059 | 7024 | 35 | 0.5% [0.33%–0.66%] |
| Cardiovascular (K) | 20,464 | 20,357 | 107 | 0.52% [0.42%–0.62%] |
| Musculoskeletal (L) | 77,854 | 77,408 | 446 | 0.57% [0.52%–0.63%] |
| Neurological (N) | 18,942 | 18,817 | 125 | 0.66% [0.54%–0.78%] |
| Psychological (P) | 39,166 | 38,875 | 291 | 0.74% [0.66%–0.83%] |
| Respiratory (R) | 53,527 | 53,162 | 365 | 0.68% [0.61%–0.75%] |
| Skin (S) | 32,719 | 32,543 | 176 | 0.54% [0.46%–0.62%] |
| Metabolic and Nutritional (T) | 20,594 | 20,485 | 109 | 0.53% [0.43%–0.63%] |

[a]The share vaccinated at least once. This rate varied considerably across the different virus types. In Supplementary Table 3 we therefore also present incidences separately among vaccinated and unvaccinated for each virus type.

Approximately similar estimates were found in analyses of our secondary outcome measures (post-COVID respiratory complaints and post-COVID fatigue; Supplementary Figs. 3, 4). In additional robustness analysis (Supplementary Fig. 5), we recoded individuals

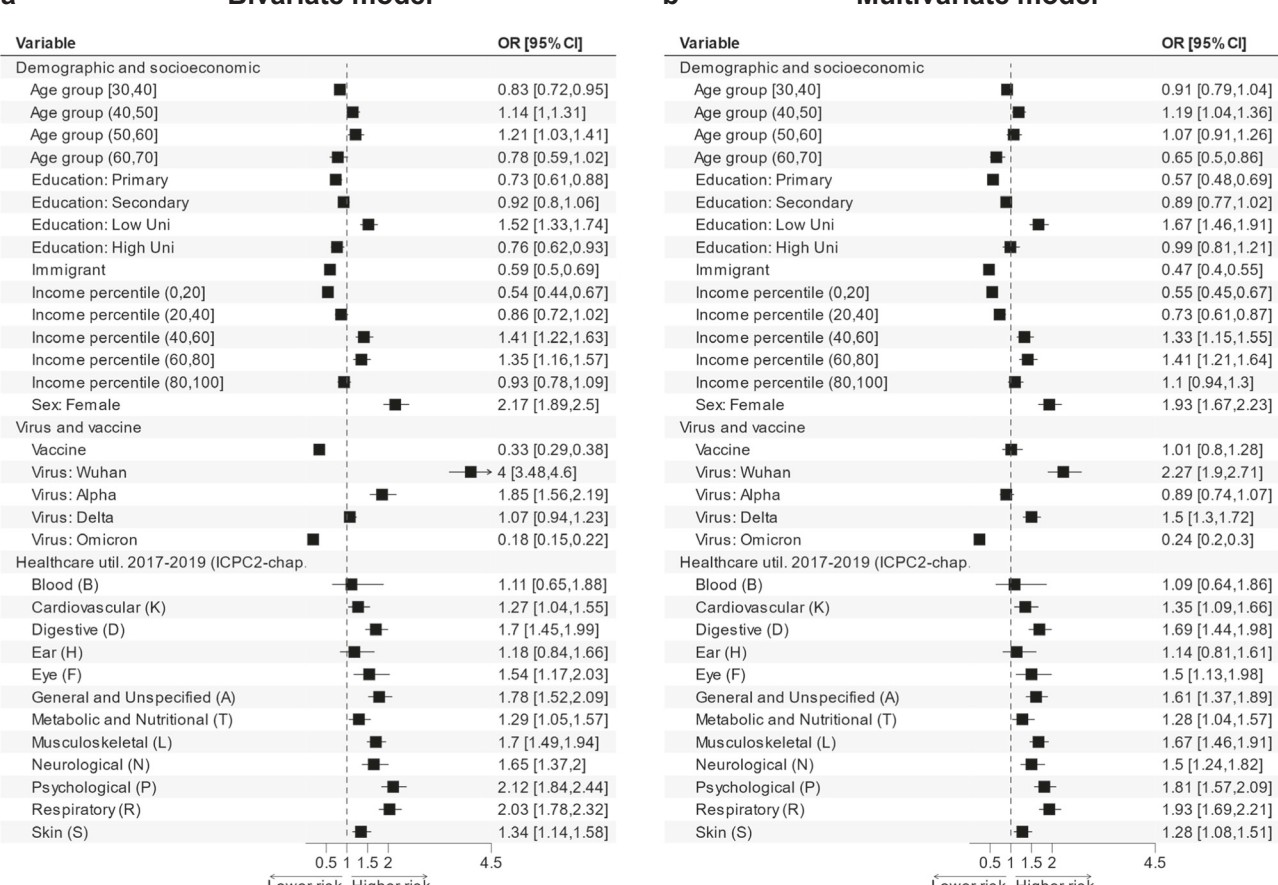

**Fig. 1 | Odds ratios for the post-COVID condition in bivariate and multivariate logistic regressions (*n* = 214,667 individuals).** The squares show the estimated ORs for the post-COVID condition for each predictor when included as a binary variable into the model, with 95% confidence intervals represented by lines. The reference group (i.e., dashed vertical line, OR = 1) for all predictors was "everyone else", i.e., everyone not having the predictor or characteristic of interest. **a** The results from bivariate models, **b** the results from multivariate models. Note that the multivariate models (**b**) used two sets of explanatory variables: for demographic characteristics, socioeconomic characteristics, virus type and vaccine status, the adjusted models accounted for health care utilization prior to the pandemic. While for health care utilization, the adjusted models accounted for demographic and socioeconomic factors, as well as vaccine status and virus type (see Table 3 for details on variable coding). Healthcare util. 2017–2019 (ICPC2-chap.) = Healthcare utilization 2017–2019 (ICPC2-chapter); Education: LowUni = Education: Lower University degree; Education: HighUni = Education: Higher University degree.

with post-COVID anxiety and depression as non-post-COVID cases, the OR for Psychological health problems was then 1.78 (95% CI 1.53–2.08).

In the supplementary material we show that our main results were robust across different sample selections: including hospitalized individuals (Supplementary Fig. 6), including individuals with reinfection within 180 days (Supplementary Fig. 7), including individuals either hospitalized and/or reinfected within 180 days (Supplementary Fig. 8) and including individuals who were infected after the initial pandemic phase (Supplementary Fig. 9). While our main analysis excluded individuals who were hospitalized due to COVID-19, it should be noted that hospitalization due to COVID-19 was the strongest overall predictor of the post-COVID condition, with an OR = 7.36 (95% CI 6.07–8.91) in the bivariate model, but reduced to aOR = 3.98 (95% CI 3.26–4.85) when controlling for pre-pandemic health care utilization.

**Prediction models**
The LASSO and random forest both had an area under the curve (AUC) of ~0.78 (95% CI 0.740–0.806 for Random Forest; 0.745–0.810 95% CI for LASSO) (Fig. 2a). This is above moderate prediction performance, reflecting that our data to a considerable extent can predict post-COVID cases. Both models perform at the same level, reflecting that little could be gained by allowing for more non-linearities (complexity)

within our set of covariates. In other words, a transparent and sparse LASSO model is sufficient with these explanatory variables.

In terms of model importance (Fig. 2b, c), the models to a large extent agreed on the most important factors for predicting future post-COVID condition. These included virus type, sex, pre-pandemic psychological and respiratory health problems, with 6 out of 10 strongest predictors being the same for both models. The models diverged somewhat with regards to the importance of vaccines. This difference likely reflects that the sparsity of the LASSO leads to the exclusion of vaccines due to its strong correlation with virus type. The Wuhan-virus was the reference category, hence the low risk for the other virus types reflects the substantially higher risk for the Wuhan-type. The sign of each predictor is also indicated in the Figure, where "POS" indicate higher likelihood of the post-COVID condition, and "NEG" a lower likelihood.

Note that the scaling of the importance scores is not directly comparable, as the LASSO scores are based on standardized coefficients, while the random forest scores are based on how prediction accuracy changes when permutating an explanatory variable.

When extending to more complex models, where we split each ICPC chapter into symptom (00–29) and diagnoses codes (30–79) and counted the number of visits instead of binary markers, the score dropped to 0.77 and 0.75 for the LASSO and random forest

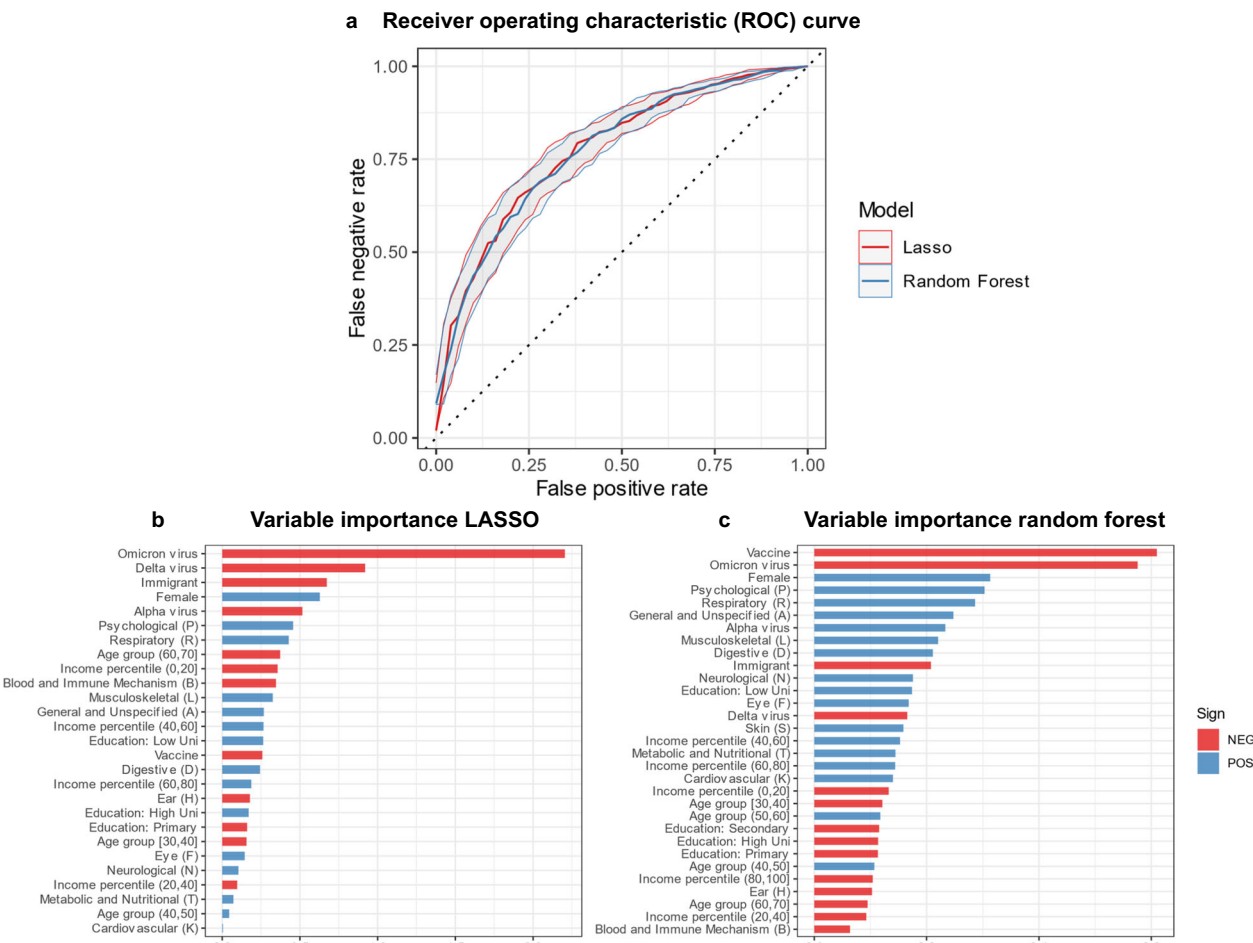

**Fig. 2 | Results from machine learning models.** Receiver operating characteristic curve (**a**), and variable importance scores (**b**, **c**).The solid lines show the ROC curve for the LASSO and random forest, in red and blue, respectively (**a**). The shaded areas indicate the 95% confidence intervals. The variable importance scores for the LASSO (**b**) were based on standardized coefficients, i.e., the beta divided by its standard error. The variable importance scores of the Random Forest were calculated by sequentially randomizing each variable and assess the drop in prediction performance (**c**). The strongest predictors are those where prediction performance drops the most when replaced with noise. The signs of the predictors in the Random Forest were determined by comparing average sample likelihoods when recoding the predictor in question on/off for all individuals. If the average sample likelihood increased, the sign was coded as "POS", otherwise "NEG". There were $n = 171{,}733$ individuals in the training set, and $n = 42{,}934$ in the test set. Education: LowUni = Education: Lower University degree; Education: HighUni = Education: Higher University degree.

respectively. For the simpler models, only trained a small set of strong predictors (virus type, vaccine status, sex, and prior psychological (P), respiratory (R) and unspecified (A) health problems), the AUC was 0.759 (95% CI 0.73–0.79)

## Discussion

In this observational study of ~214,000 individuals with confirmed COVID-19, we found that pre-pandemic health care utilization related to psychological, respiratory, and general/unspecified health problems were the strongest predictors for having a doctor-diagnosed post-COVID condition between 90 and 180 days after the initial infection. We also found that women, and individuals infected by the original (first) virus variant, had a higher risk of post-COVID complaints. There was no strong or clear social gradient in the prevalence of the post-covid. Last, we found that when accounting for virus type, vaccination was not significantly associated with the post-COVID condition.

### Comparison to previous studies

This study found a 0.42% incidence of post-COVID complaints among those infected with SARS-CoV-2 between July 1st 2020 and January 24th 2022. This is low compared to several previous survey-based studies, which sometimes report an incidence above 10%. However, it aligns well with a recent study from the UK utilizing electronic health records from primary care which found a prevalence of 0.4%[6]. Such large differences in incidence likely reflects that post-COVID cases identified using health care utilization data could be considered as more severe than those measured in surveys. In summary, the incidence of post-COVID complaints varies considerably depending on measurement methods[7].

The higher incidence among women and middle-aged (i.e., declining at higher ages) found in our study has also been observed in several other studies[6,12,22]. We found indications of a U-shaped association between income and the post-COVID condition, i.e., that individuals with middle income (40th to 80th percentile) has higher odds for having the post-COVID condition than other individuals. Moreover, individuals with low university education had a higher odds of the post-COVID condition when compared to individuals with other education levels. These findings of socioeconomic gradient are partly contradictory to findings reported in other studies[6,21]. It should be noted, however, that the absolute differences in resources between the top and bottom of socioeconomic distributions differ significantly between countries, making cross-country comparisons difficult to interpret.

To our knowledge, the increased risk of post-COVID complaints following infection by the original virus variant, compared to the other virus types has not been shown before; while the lack of protective effect of vaccines (Fig. 1, panel b) is in line with previous studies suggesting no, or only partial protection[23–25].

Regarding pre-pandemic healthcare utilization, the increased risk among individuals with respiratory health problems or adverse prior mental health is comparable to what was found in recent studies[6,12,21]. The repeated finding of a positive association between post-COVID complaints and poor mental health in our and previous studies raises important and difficult questions related to underlying mechanisms. We suggest these associations as topics for future studies[20]. The increased risk among individuals with general and unspecified health problems has, to our knowledge, not been shown in primary healthcare data. This heightened risk could reflect individual differences in help-seeking when experiencing health problems after a COVID-19 disease.

The prediction performance from the machine learning models reflects that pre-infection data is informative about the likelihood of post-COVID. For comparison, a recent study based on data collection through an app (The COVID Symptom Study), showed an AUC = 0.76[12]. However, contrary to our study, this study used the post-infection measures—number of symptoms reported within the first week—as predictors, and duration beyond 28 days as a post-COVID case. It also worth noting that in our case two models with very different degrees of complexity had similar AUCs, suggesting that a simple (linear model) and sparse (few explanatory variables) model would perform almost equally well.

### Interpretation and relevance

We found that a limited set of predictors provided substantial information regarding the risk of post-COVID complaints. To illustrate, while the average risk in our sample was 0.42%, the risk among those with the strongest risk factors was 10-fold (~4%; original virus, female, prior history (2017–2019) of respiratory, psychological and general health problems). These findings imply that a simple checklist of yes/no questions may function as a prognostic tool for predicting post-COVID health complaints. As such, we believe our study has important implications for care providers, for example in general practitioner settings and/or in specialist care settings. When our findings are validated in other samples and populations, a checklist of yes/no question as described above can be implemented into clinical practice to provide information about a patient's prognosis following COVID-19. Such knowledge may be important for timely treatment decisions and/or for prevention of long-term sickleave (at least when the same doctor is following the same patient over time and when the same doctor is responsible for prescribing sick leave). The checklist may further provide important knowledge for rehabilitation personnel. For example, when a patient has undergone COVID-19 and receives physiotherapy in the recovery period, the interventions could be better targeted to the individual based on the knowledge that the individual is at no risk or at enlarged risk of the post-covid condition. Last, when validated out of sample, we believe our checklist may also be used in the selection of individuals to future trials or observational studies. However, it should be noted that not everyone with a positive test will visit primary care with complaints, and treatment options are currently limited. We have previously reported that the prevalence of common medical complaints and health care visits following COVID-19 is elevated particularly 1–3 months after positive test[26,27]. A small proportion of the individuals visiting primary care during 1–3 months post COVID will still need care at 4–6 months, however it is unclear what care would be helpful for this group of individuals. As such, the proposed checklist may be useful among individuals testing positive who are symptomatic, i.e., individuals visiting their doctor with complaints in the acute and/or sub-acute COVID-19 phase, when more treatment options are available. We believe this potential clinical usefulness of our findings as well as timely treatment options should be further investigated in future studies.

### Strengths and weaknesses

The main strength of the study is that it covers close to all infected individuals in Norway during the study period, as well as complete registrations of their health care utilization in the years prior to the pandemic. Hence, the study to a very limited degree suffers from sample selection or recall bias. Moreover, healthcare utilization and socioeconomic characteristics were measured prior to the pandemic, hence there is no risk of reverse causation.

The study has several limitations. Our main limitation is that the outcome measure was constructed based on health care utilization. There could be considerable heterogeneities across the population with regards to the propensity to seek help. Hence, the prevalence measured along any given dimension—e.g., age, sex or socioeconomic background—could entail considerable bias, to the extent that these dimensions correlate with health-seeking behavior. For example, the higher prevalence found among women and middle-aged could potentially be explained by a higher inclination to seek professional help for health problems in these groups.

Another potential limitation of our study is the risk of misclassification related to whether general practitioners accurately reported positive COVID-19 tests to the official register. Although general practitioners were required by law to report all cases of COVID-19 to the official register, there could exist cases where infections were not reported, potentially resulting in false post-COVID classification (i.e., the post-covid condition as reported in our study could be due to a second episode of COVID-19). However, we consider this risk limited, as measures were put in place to remind general practitioners of the immense importance of continuously reporting cases during a national health crisis. Further, reinfection was rare: ~2% of those infected in between July 2020 and Aug 2022 were infected twice within 180 days (Supplementary Fig. 1).

Lastly, the post-COVID condition was a new phenomenon in the early phases of the pandemic and general practitioners may not have known how to interpret, or code, the symptoms reported by their patients. Although it was possible to register an R992 code together with a persistent complaint, the primary care physicians might not have done so. The operationalization chosen in this study is in line with the official guide given to general practitioners in April 2021 and in accordance with the WHO definition of the post-COVID condition. We found similar results in our sensitivity analysis where inclusion was started in January 2021, with corresponding potential post-COVID cases from April 2021 (Supplementary Fig. 9). Still, there may be individual variations in coding practices, which might have influenced results.

Moreover, patient groups prefer self-diagnosis as a definition, and we believe our findings need to be replicated and/or nuanced in future studies using patient-reported outcome measures. Along this line, there may be important post-covid complaints not studied here. For example, loss of taste and smell are commonly reported among patients[1,28] but could not be studied here because of low numbers.

In conclusion, we found that individuals with mild initial COVID-19 and a prior history of psychological, respiratory, or unspecified/general health problems, had a higher risk of developing post-COVID complaints. There was also an increased risk among women and those infected by the Wuhan-virus. When validated in other samples and settings, these findings may be used by clinicians and care providers to inform about the prognosis after COVID-19 regarding the development of the post-covid condition.

## Methods
### Study design and participants
Using a prospective cohort study design following individuals for up to 180 days after the first positive test, we utilized data from the

Norwegian Emergency Preparedness Register, Beredt C19 (BC19). BC19 is a national database containing linked register data aiming to provide rapid knowledge to authorities in handling the pandemic. Sources included in the current study were the Norwegian Population Register (demographic characteristics), the Norwegian Tax Authorities and National Education Database (socioeconomic variables), the Norwegian Surveillance System of Communicable Diseases (results from all Polymerase Chain Reaction (PCR) testing), the Norwegian Immunization Registry (data on all vaccination against COVID-19), the Norway Control and Payment of Health Reimbursement Registry (primary health care visits before and during the pandemic) and the Norwegian Patient Registry (specialist health care visits before and during the pandemic). These data sources were linked using a deidentified version of the personal identification number received upon birth or immigration.

Our study population included all Norwegian residents aged between 30 and 70 years old (i.e., working age individuals) on Jan 1st 2020, and who had their first positive SARS-CoV-2 PCR test, as registered in the Norwegian Surveillance System of Communicable Diseases, between July 1st 2020 and January 23rd 2022. By including individuals from their date of first positive test, we could ensure that the included individuals had no pre-existing post-covid complaints resulting from previous COVID-19 illness.

We excluded individuals with one or more positive tests in the period 31 to 180 days after the first positive test. In this way, we could exclude new onset symptoms that were due to a new SARS-CoV-2 infection and not related to the first SARS-CoV-2 infection (i.e., all positive test occurring the first 30 days after the first positive test were regarded to result from the same infection period[29]). We also excluded individuals that were hospitalized due to COVID-19 as these experienced considerably more bodily stress from the infection. We required complete follow-up data, i.e., all individuals were followed for 180 days after testing positive.

### Outcome: post-COVID condition

The main outcome of interest was having the post-COVID condition (yes/no) as recorded by a general practitioner (GP) in primary or emergency care by the International Classification of Primary Care code (ICPC-2). From May 4th 2020, primary care physicians were instructed to use the code R992 diagnosis for patients with COVID-19 disease. Persistent complaints after COVID-19 were coded by an R992 code together with at least one code for a persistent symptom, for example fatigue or pain[30]. For example, if a patient reported to be struggling with fatigue after the infection, it was coded with R992 together with A04 (weakness/tiredness). Correspondingly, if the complaint was shortness of breath, it was coded with R992 and R02. This coding for persistent complaints was possible for primary care physicians to use at any time during the pandemic. However, an official recommendation to do so was provided by national health authorities from April 1st 2021. The recommendation stated that persistent COVID-19 complaints should be coded by primary care physicians based on patient history of persistent complaints and an earlier, confirmed COVID-19. In our study, we assessed physician-reported post-COVID condition for one or more of several long-term symptoms after a SARS-CoV2 infection as described in Table 2[31], if they occurred in the time range 90-180 days after the first positive test. As such, our definition is in accordance with the World Health Organization's definition of post-covid conditions (covid-like complaints present 3 months after infection)[8]. The assumption of our main outcome "post-COVID condition" was that the risk of the diverse symptoms together makes up the risk of the post-COVID condition, which we assume shares common predictors. However, the predictors may differ by symptoms, and to examine the sensitivity of our results we also assessed two secondary outcome measures, based on findings in previous register-based research[27,32] and the number of observations for each outcome

**Table 2 | Diagnostic codes of conditions/complaints[a] used in concurrence with "R992" (confirmed COVID) to operationalize the post-COVID condition (ICPC-2)**

| Description of health problem | ICPC-2 code |
|---|---|
| Pain (general/multisite and localized pain and symptoms from the musculoskeletal system, not classified elsewhere (neck, back, arms/hands, feet/legs)) | A01, L01-L17, L18-L20, L29 |
| Fatigue | A04, A05, A29 |
| Cough | R05 |
| Heart palpitations | K04, K05, K29 |
| Shortness of breath | R02 |
| Anxiety and depression | P03, P76, P01, P74 |
| Brain fog (concentration or memory problems) | P20 |

[a]With condition/complaint we refer to all information that may be included in an ICPC-2 (International Classification of Primary Care 2) code: Diseases, disorders, signs, symptoms, and/or complaints as classified by the physician consulted.

in our sample: (1) Respiratory complaints (including cough and shortness of breath) and (2) fatigue (Table 2). As a robustness check, because individuals with anxiety and/or depression might be more prone than others to seek medical care due to health concerns also for physical health issues[33], we also examined how the results were affected when recoding individuals with anxiety and depression post-COVID symptoms as non-post-COVID cases.

Medical recording to the National registries is mandated by law in Norway, reducing potential bias due to missing data in our study. Norwegian health register data have been demonstrated to have high validity and reliability in a small comparative study of medical journal notes and medical records[34], i.e., they are well suited for studying patterns of health care use and complaints leading to health care use. Still, we made use of a diagnostic coding practice that was introduced during the pandemic and therefore was relatively new to primary care physicians. Indeed, the use of the codes as described above was limited in the beginning of the pandemic (when both the post-COVID condition was new, and also coding practices were new), before slowly rising and reaching its top in March 2022 (Supplementary Fig. 10).

### Predictors

We included predictors based on demographic and socioeconomic characteristics, previous healthcare use, virus variant, and vaccination against COVID-19 (Table 3), all as identified by the registries described above. For "health care utilization prior to infection" (Table 3), we relied on the pre-pandemic period 2017-19 because of periodically restricted access to care during the COVID-19 pandemic and hence corresponding differences in the data generating process during the different phases of the pandemic. Virus variant was identified based on which virus type was dominant among infected individuals: the Wuhan virus (March 1st 2020– February 16th 2021), the Alpha virus (February 17th 2021–June 30th 2021), the Delta virus (July 1st 2021–December 23rd 2021), and the Omicron virus (December 24th 2021–January 23rd 2022).

### Statistical analyses

The statistical analysis consisted of two parts. In the first part we explored the incidence and risk factors for doctor-diagnosed post-COVID condition. In the second part we built prediction models using machine learning algorithms.

We estimated the incidence of post-COVID condition for each stratum of the included covariates (Table 3). We then estimated Odds Ratios (OR) for the post-COVID condition in bivariate and multivariate logistic regression models. While the bivariate models only contained the outcome and exposure of interest (each factor separately), the multivariate models used two different sets of explanatory variables,

**Table 3 | Overview of predictors used in the analysis**

| Demographic variables | |
|---|---|
| Gender | Binary variable; coded as 1 for women. |
| Age | Categorical variable for age groups 30–39;40–49;50–59;60–69. |
| Immigrant | Binary indicator for whether the individual was born in Norway |

| Socioeconomic variables | |
|---|---|
| Education | The highest level of education achieved during life, divided in 4 categories: Primary education (ISCED-11 levels 1–3), secondary education (ISCED-11 levels 4–5), low university education (ISCED-11 level 6), high university education (ISCED-11 levels 7–8). |
| Income | Birth cohort- and gender-stratified income quintile, i.e., 5 categories based on the individual annual income. |

| Health care utilization prior to infection | |
|---|---|
| Primary care consultations | For each chapter in the International Classification of Primary Care (ICPC-2) coding system we created a categorical variable indicating whether the individual had one or more registered consultations in the period 2017–2019. |

| Vaccination status and virus type | |
|---|---|
| Vaccine | Binary indicator for whether the individual was vaccinated before infection. |
| Virus type | Wuhan-Hu-1 (hereafter "Wuhan-virus"), Alpha-virus, Delta-virus or Omicron-virus. Coded as a binary indicators based on the month when the individual was infected. The assigned virus type was the dominant type in the month of infection. |

depending on which factor was under study: (i) When analyzing healthcare utilization, we controlled for all the demographic and socioeconomic factors and vaccination status. (ii) When studying the risk related to demographic and sociodemographic characteristics, and vaccine status, we ran separate models for each characteristic while adjusting for the healthcare utilization prior to infection (2017–2019). To illustrate, the adjusted model for a specific age group shows the risk adjusted for health care utilization history. Note that since virus type and vaccination status at the time of infection were strongly correlated, the multivariate models analyzing virus types included both vaccination status and sociodemographic factors as controls. We repeated our analyses for the secondary outcome measures (post-COVID respiratory complaints and post-COVID fatigue) and when recoding individuals with anxiety and depression post-COVID symptoms as non-post-COVID cases. We also repeated the main analyses in several sensitivity analyses related to the study sample: (1) An analysis of risk factors when including hospitalized individuals, (2) an analysis of individuals with reinfection within 180 days, (3) an analysis of individuals either hospitalized and/or reinfected within 180 days, (4) an analysis of individuals who were infected after December 2020 (as opposed to the first period when the virus and its short- and long-term consequences were unknown). For a more standardized interpretation of predictor-specific incidence and odds ratios, we used "everyone else" as the reference group in all analyses. Thus, all predictors were added to the model as a binary 0/1 variable, where 1 represented having the characteristic of interest (for example Age group (50,60]) taking value 1), and 0 represented everyone else, not having the characteristic of interest (in the example, all other age groups, i.e., age groups [30,40], [40,50], [60,70] taking value 0). Likewise, for predictor Female, coded as 1, everyone else, who were typically categorized as Male, were coded as 0. As such, the odds ratio for females will be the inverse of the odds ratio for males and vice versa.

The aim of the machine learning models was to predict post-COVID cases. We built prediction models with two different machine learning algorithms, one transparent (Least Absolute Shrinkage and Selection Operator, or LASSO) and one more flexible and opaque (Random Forest). To limit overfitting, both models were tuned with cross-validation.

The Least Absolute Shrinkage and Selection Operator (LASSO) is one of several penalized regression methods available for prediction[35]. Due to its sparsity and performance, the LASSO has become widely used when aiming for an interpretable, yet well performing out of sample, predictive model. What separates the LASSO from other penalized regression models is the functional form of the penalty term: The LASSO uses the absolute sum of coefficients (L1 penalty), while other methods use the sum of squared coefficients (L2 – ridge regression), or a combination of both (elastic net regression). The result is that the LASSO tends to suggest sparse models, keeping only a small set of strong predictors.

The random forest averages the predictions from multiple Classification and Regression trees (CARTs)[36]. Hence, it is an "ensemble learner". The random forest has gained popularity due to its high level of performance, robustness to various data challenges (missing observations, rescaling of predictors etc.) and limited set of tunable hyperparameters. What is particular with the random forest is that each CART is fit using only a random subset of the available predictors. This random selection of predictors has been shown to boost the predictive performance by limiting inefficient dependency across individual CARTs.

Both the LASSO and random forest models were estimated on the full set of covariates, i.e., both sociodemographic data, health care utilization data, vaccine status and virus type. The outcome was a binary indicator of the post-COVID condition. The models were tuned using 10-fold cross validation with the same folds across model types, and performance was assessed on the same hold-out sample (20%). Using bootstrapping, we also estimated the confidence intervals of their performance (area under the curve).

We extended our machine learning models in two directions. First, to explore the potential for improving the AUC score by adding complexity, we also estimated models where we split each ICPC chapter into symptom (00–29) and diagnoses codes (30–79). Moreover, instead of a binary marker for primary healthcare we counted the number of visits for each symptom and diagnosis. Second, to explore the potential for simplifying the model in order to make it more clinically relevant, we estimated models which only a small set of strong predictors.

All prediction analysis was done within the Tidymodels machine learning framework. The confidence intervals for the area under the curve were estimated with the pROC-package, using the Delong-method[37]. All analyses were run in R (v.4.0.2), using the packages tidyverse (v.1.3.2), broom (v.1.0.2), tidymodels (v.0.1.4), ranger (v.0.13.1), and glmnet (v.4.1–3). The data from the different registers were linked in R using the RODBC-package (v.1.3-19).

### Inclusion and ethics
The Ethics Committee of South-East Norway confirmed on June 4, 2020 that external ethical board review was not required (#153204).

The data sources (The emergency preparedness register for COVID-19) were established and handled in accordance with the Health Preparedness Act §2-4 (11), enabling a quick and responsive way for the Norwegian government to access knowledge of how to handle the pandemic. Hence, the data and analysis were regarded by the ethical committee to respond to research aims not falling under the Law of Health Research §§ 2 and 4a. Informed consent from participants was not required, since the study was based on routinely collected administrative register data. Data from the different registers were linked by the certified researchers and using an encrypted personal ID-variable. Unencrypted ID-numbers were not available to the researchers. All methods were carried out in accordance with relevant guidelines and regulations. To protect participant privacy and security of personal data, all data were handled under strict confidentiality and access control as described in the Norwegian Institute of Public Health's internal documentation.

### Reporting summary

Further information on research design is available in the Nature Portfolio Reporting Summary linked to this article.

## Data availability

The study was based on the Emergency Preparedness Register for COVID-19 (Beredt C19), a strictly regulated register available to selected authorized researchers in Norwegian Institute of Public Health. The individual-level data that support the findings is thus not publicly available due to privacy laws. However, the data are accessible to authorized researchers after ethical approval and application to "helsedata.no/en" administered by the Norwegian Directorate of eHealth. The response time for data applications, following the necessary ethical approvals, varies by the demand and capacity at each register. It can range from months to two years, depending on the circumstances.

## Code availability

Code used for producing the results presented in this study is available at https://github.com/remebjornatle/post_covid or https://zenodo.org/badge/latestdoi/661041995[38].

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

## Acknowledgements

This study was supported by the Norwegian Institute of Public Health (internal funding, no grant number available). We thank the Norwegian Directorate of Health, particularly Olav Isak Sjøflot and his Department of Health Registries for their cooperation in establishing the emergency preparedness register, and Gutorm Høgåsen and Anja Elsrud Schou Lindman for their invaluable work on the register. The interpretation and reporting of the data are the sole responsibility of the authors, and no endorsement by the register is intended or should be inferred.

## Author contributions

B.-A.R. and J.G. performed the statistical analyses and drafted the manuscript. B.-A.R., J.G. and K.M. contributed to the research design. All authors (B.-A.R., J.G. and K.M) critically reviewed the manuscript for important intellectual content. All authors (B.-A.R., J.G. and K.M.) gave final approval for the version to be submitted.

## Competing interests

The authors declare no competing interests.
