## [Peer Review File · Nature Communications]

Predictors of the post-COVID condition following mild SARS-CoV-2 infectionREVIEWER COMMENTS

Reviewer #1 (Remarks to the Author):

Thanks for inviting me to review this interesting paper. The strengths and uniqueness of this paper include a high-quality complete follow-up data up to 180 days of COVID infection, thoughtfully constructed cohort that excluded cases with potential confounders, and appropriate statistical techniques that were efficient and effective in identifying the most predictive factors for the risk of post-COVID conditions.

My major concern is about the outcome of this study. The authors created a single dichotomous outcome of the post-COVID conditions by using the "or" logic to combine seven categories of diagnosis (pain, fatigue, cough, heat palpitations, shortness of breath, anxiety and depression, and brain fog). What was the assumption for this outcome? Did the authors assume that the risk of such diverse conditions shared common predictors? I would like to ask the authors to provide the frequency of each condition/complaint in Table 1 to help people understand the most/least prevalent post-COVID conditions. Ideally, I would recommend the authors modeling each of the seven conditions/complaints in Table 1 as a separate outcome. If the prevalence of a given condition is too low, then the authors may consider creating outcomes by combining conditions of the same organ systems (e.g., pulmonary combining cough and shortness of breath). At least, I think the authors should create separate outcomes for risks of physical (pain, fatigue, cough, heat palpitations, and shortness of breath) and cognitive (anxiety and depression, and brain fog) conditions.

As reported, the COVID infection dates ranged from 07/2020 through 01/2022 and the period of history health care utilization ranged between 2017 and 2019. For patients infected in 01/2022 (Omicron?), all their history records were at least 2 years ago. I would ask the authors to provide the distribution of intervals (in months) between the COVID infection and history healthcare-utilization among patients. If the intervals were widely different among patients, please evaluate/estimate potential impact/limitation of adjusting for these history factors in modeling.

In Table 3, viral variants should not be listed under "Healthcare utilization 2017-2019". Also, please provide the date range that were used to identify each of the four viral variants (Wuhan, Alpha, Delta, and Omicron).

Reviewer #2 (Remarks to the Author):

This study uses data on the Norwegian population to try to predict the likelihood of long-Covid following acute infection. I cannot comment on machine learning as I don't have experience of this methodology. I do, however, have other comments and concerns.

Major issues

- A stronger argument needs to be made for the practical implications of this work, especially given that there is little evidence for effective treatment for long-Covid symptoms (pacing being one). I fail to see how predicting the likelihood of long-Covid will "prevent long-term illness, sick leave, and disability".
- Likewise, it seems unrealistic that a checklist might be used to predict the prognosis of individuals with Covid-19. Most people do not consult a doctor (indeed medical professionals would not wish to see patients with mild infection).
- How well is long-Covid diagnosed by doctors and recorded in Norway? Were patients believed from early in the pandemic? How has prevalence/recording changed over time? Patient groups prefer self-diagnosis as a definition.
- According to the flow chart 8.9% of the population aged 30-70 were infected with Covid-19 in a two year period early in the pandemic. Even if you allow for repeat infections this does not tally with WHO figures for Norway <https://covid19.who.int/region/euro/country/no>

Minor issues/points of clarification

- Why was the age range limited to 30-70 years?
- Was follow-up limited to 180 days?
- Individuals who were hospitalised were excluded. Therefore, this is a study of mild infection. This should be made clear in the title, abstract, and conclusions.
- Table 1 only lists some symptoms of long-Covid. If these are the only ones under consideration it should be listed as a limitation. How about others, for example headache?
- Page 4 "Medical recording to the National registries is mandated by law in Norway, ensuring no missing data in our study". No missingness cannot be guaranteed. There will be individual variation in diagnostic practice, which should be mentioned in the discussion.
- Are there any data on ethnicity? A binary immigrant/non-immigrant variable is crude. Immigrants will be a heterogeneous group.
- Is income recorded at the individual or household level? Both measures will have a level of error.
- Is number of primary care consultations a binary variable, or are there more categories? There is a big difference between consulting primary care once over a period of a few years, versus regularly.
- Please add reference groups to all odds ratio plots. It does not make sense to have separate odds ratios for (for example) males and females. One should be the referent unless I am misunderstanding the analysis.
- Typo on page 2 "16% higher risk among those most socioeconomically deprived" should be 11%.
- Typo on page 9 "had a psychological diagnosis were 121 percent more likely" should be 12%.

Reviewer #3 (Remarks to the Author):

Thank you for asking me to review this paper. The authors use data from a national Norwegian registry, identifying people who tested PCR positive for COVID-19 between July 2020 and Jan 2022 and analysing socioeconomic, demographic, vaccination and healthcare-utilisation data to look for predictors of post-COVID condition. They identify several factors that have been well established in other studies (female sex, vaccination status, COVID variant) and the authors focus on the findings relating to comorbidities, which show elevated risk for a number of conditions (as identified by pre-covid healthcare use).

I have some reservations about the findings and these are mostly related to the definition of the outcome. The binary post-covid condition outcome is defined as a ICD-10 code of R992 (confirmed COVID-19) plus any one of a range of documented conditions (fatigue, cough, palpitations, shortness of breath, anxiety, depression, brain fog, musculoskeletal pain). What is unclear is how many of these conditions were pre-existing, and the extent to which these conditions can be confidently linked to the COVID-19 infection. Clearly it is not a requirement for the patient not to have experienced the condition before, as pre-existing anxiety and depression are identified as risk factors. Is it down to the physician's judgement as to whether the condition has been exacerbated by COVID? Whatever the answer, this needs some clarification and ideally some more interrogation. Otherwise it might be argued that you're concluding that eg having depression before COVID is associated with having depression after COVID.

My suggestions would be:

- > Include a paragraph explaining in more detail what the criteria were for the physicians to associate the health problems with COVID-19 infection.
- > Include a table showing a breakdown of coded post-covid health problems among the 908 people with post-covid condition. Ideally, also show how many of these people had these conditions before their COVID-19 diagnosis.
- > A useful sensitivity analysis would be to remove, for example, people with anxiety and depression related post-covid condition and see if pre-existing anxiety and depression are still identified as risk factors.

Some other comments:

> The methods describe logistic regression models and odds ratios but the figure legends refer to Cox models.

> The phrasing of the specification of the adjusted logistic regression models was a little confusing.

“When studying demographic, sociodemographic and vaccine status information, the multivariate models adjusted for healthcare utilization prior to infection”

- does this mean that they *only* adjust on healthcare utilization? Or that they adjust on healthcare utilization *and* sociodemographic/vaccine variables?

> How were the signs on the RF variable importance determined?

> In the bivariate analyses, what were the reference categories? For instance, in the income categories, are you comparing 80-100 percentile vs pooled 0-80 percentiles? If so, I think it would be more interpretable to regress all percentile categories together as a single categorical variable and choose a reference category.

> Relating to the above, in the discussion you say that you observed no clear socioeconomic gradient. However, it does seem like there is a gradient, especially in income. The effect might be more apparent in a model with a categorical predictor.

We would like to thank the editorial board and reviewers for having performed a careful review and consideration of our study, which we think has greatly contributed to further enhance its quality. The reviewers raised several important concerns, and there were many good suggestions of how to improve our work that led us to revise accordingly. While we have made several changes (see below for details), we have particularly focused on 1) sensitivity analyses with regards to the outcome variable, and 2) further discussion of the implications of our work. During the process of quality control, we also discovered that Figure 2 in the submitted version had not been updated correctly from a prior version. We have now corrected this mistake.

Please see the detailed point-to-point responses and actions to the editors’ comments beneath. Page references refer to the marked version of the manuscript.

#	Reviewer 1	Our response	Action
1	Thanks for inviting me to review this interesting paper. The strengths and uniqueness of this paper include a high-quality complete follow-up data up to 180 days of COVID infection, thoughtfully constructed cohort that excluded cases with potential confounders, and appropriate statistical techniques that were efficient and effective in identifying the most predictive factors for the risk of post-COVID conditions.	Thank you for a positive evaluation.	
2	My major concern is about the outcome of this study. The authors created a single dichotomous outcome of the post-COVID conditions by using the “or” logic to combine seven categories of diagnosis (pain, fatigue, cough, heat palpitations, shortness of breath, anxiety and depression, and brain fog). What was the assumption for this outcome? Did the authors assume that the risk of such diverse conditions shared common predictors? I would like to ask the authors to provide the frequency of each condition/complaint in Table 1 to help people understand the most/least prevalent post-COVID conditions. Ideally, I would recommend the authors modeling each of the seven conditions/complaints in Table 1 as a separate outcome. If the prevalence of a given condition is too low, then the authors may consider creating outcomes by combining conditions of the same organ systems (e.g., pulmonary combining cough and shortness of breath). At least, I think the authors should create separate outcomes for risks of physical (pain, fatigue, cough, heat palpitations, and shortness of breath) and cognitive (anxiety and depression, and brain fog) conditions.	Thank you for your thoughtful comment regarding our outcome measure. We agree that more detail is needed on our choice of operationalization of the post-COVID condition. The assumption for this outcome was indeed that the risk of the diverse conditions together makes up the risk of the post-covid condition, which we assume shares common predictors. We acknowledge the need for clearer communication of this assumption in the Methods section. The conditions chosen were based on our previous own register-based research (Magnusson et al., Nat Comm 2023) as well as a previous large cohort study of post-COVID complaints (Caspersen et al., 2022). We agree that a frequency table providing the frequency of each condition/complaint in Table 1 would be useful and provide more transparency to our work. Further, given the very detailed predictor data, we believe that any added prediction analyses of added outcomes should be based on a combined judgement of numbers having the outcome and clinical relevance.	We have provided more detail on the assumptions of our outcome measures and as added analyses of secondary outcome measures, please see Methods, p. 4: “The assumption of our main outcome “post-COVID condition“ was that the risk of the diverse symptoms together makes up the risk of the post-COVID condition, which we assume shares common predictors. However, the predictors may differ by symptoms, and to examine the sensitivity of our results we also assessed two secondary outcome measures, based on findings in previous register based research ^{26,27} and the number of observations for each outcome in our sample: 1) Respiratory complaints (including cough and shortness of breath) and 2) fatigue (see Table 1). As a robustness check, because individuals with anxiety and/or depression might be more prone than others to seek medical care due to health concerns also for physical health issues²⁸, we also examined how the results were affected when recoding individuals with anxiety and depression post-COVID symptoms as non-post-COVID cases.” Further, we have added a frequency table showing the frequency of each of the conditions that were included in the post-covid condition in Supplementary Table 1, referred to in the results section (p. 9). We have also added frequencies in text for the most common diagnoses that were grouped together in additional analysis of secondary outcomes, with a reference to the table in the results section, p.8-9: “In total, 0.42% (N = 908) were classified as having a post-COVID condition (main outcome). Among these participants, 206 (21%) were classified as

			experiencing post-COVID respiratory complaints, while 584 (60%) were classified as experiencing post-COVID fatigue (cf. Supplementary Table 1). Of the 206 individuals with post-COVID respiratory complaints, 191 (93%) were new onsets compared to the period between 2017 and 2019 (Supplementary Table 1). Similarly, out of the 584 individuals with fatigue, 444 (76%) had new onsets of fatigue complaints. Hence, for most participants these complaints were more likely to be due to the infection and not already preexisting conditions.”
3	As reported, the COVID infection dates ranged from 07/2020 through 01/2022 and the period of history health care utilization ranged between 2017 and 2019. For patients infected in 01/2022 (Omicron?), all their history records were at least 2 years ago. I would ask the authors to provide the distribution of intervals (in months) between the COVID infection and history healthcare-utilization among patients. If the intervals were widely different among patients, please evaluate/estimate potential impact/limitation of adjusting for these history factors in modeling.	Thank you for bringing up this important topic, we agree our choices raise questions. During the COVID-19 pandemic, access to primary and specialist healthcare was periodically severely restricted, preventing patients to visit their doctor and to have their real healthcare needs recorded and thus observable in our data. If we would have chosen pre health care use for example the last year or two years prior to infection as a measure of pre healthcare use for each individual, we would have introduced systematical differences in pre healthcare use which were not due to differences in underlying health, but rather due to differences in access to care. Thus, the limited access to healthcare during COVID-19, and the how this access varied based on infection levels caused us to rely on healthcare utilization measured prior to the pandemic. We agree that providing the distribution of intervals between infection and history healthcare use is useful and that it should be evaluated.	We have added a reason for our choice of pre-pandemic period to the methods section at p. 5: “For “health care utilization prior to infection” (Table 2), we relied on the pre-pandemic period 2017-19 because of periodically restricted access to care during the COVID-19 pandemic and hence corresponding differences in the data generating process during the different phases of the pandemic.” To the results section, we have added the following with reference to a supplementary figure describing the distribution of intervals, p. 9: “Supplementary Figure 3 shows that the majority of the included individuals had their pre infection healthcare use measured approximately two years ago, and a smaller part had it measured zero to one year ago. Descriptive characteristics by pandemic period (based on virus dominance) showed that the pre healthcare use was balanced across groups, i.e., not dependent on the time interval passing between the date of SARS-CoV-2 infection and registration of previous complaints/healthcare use (Supplementary Table 2).”
4	In Table 3, viral variants should not be listed under “Healthcare utilization 2017-2019”. Also, please provide the date range that were used to identify each of the four viral variants (Wuhan, Alpha, Delta, and Omicron).	Thank you for noticing. We agree.	We have changed the table (and all figures) such that vaccine and virus are a separate category from healthcare utilization and demographic and socioeconomic characteristics. We have also added the following to the methods section, p. 5: “Virus variant was identified based on which virus type was dominant among infected individuals: the Wuhan virus (March 1st 2020 – February 16th 2021), the Alpha virus (February 17th 2021 – June 30th 2021), the Delta virus (July 1st 2021 – December 23th 2021), and the Omicron virus (December 24th 2021 – January 23rd 2022).”
#	Reviewer 2	Our response	Action
1	This study uses data on the Norwegian population to try to predict the likelihood of long-Covid following acute infection. I cannot comment on machine learning as I don’t have experience of this methodology. I do, however, have other comments and concerns.	Thank you.	

2	Major issues - A stronger argument needs to be made for the practical implications of this work, especially given that there is little evidence for effective treatment for long-Covid symptoms (pacing being one). I fail to see how predicting the likelihood of long-Covid will “prevent long-term illness, sick leave, and disability”.	We agree.	We have added the following to the introduction section, p. 2: “For example, knowing upfront that an individual with COVID-19 is at heightened risk of post-COVID condition may aid clinicians to take early action to limit long-term consequences, e.g., through early referral to rehabilitation.” In addition, we think this issue can be better described in the discussion section, please see response and action to the next comment (Reviewer 2’s comment #3).
3	- Likewise, it seems unrealistic that a checklist might be used to predict the prognosis of individuals with Covid-19. Most people do not consult a doctor (indeed medical professionals would not wish to see patients with mild infection).	We agree that the proposed checklist warrants further explanation and that some of our statements in the discussion section should be moderated.	We have added the following to the discussion section: First, we have moderated our claim of a checklist being a “powerful tool” into a “checklist may function as tool”, p. 16: “These findings imply that a simple checklist of yes/no questions may function as a prognostic tool for predicting post-COVID health complaints.” Second, we provide more information on how a checklist may function, given the Norwegian healthcare system where GPs are responsible for prescribing sick leave, p. 16: “Such knowledge may be important for timely treatment decisions and/or for prevention of long-term sickleave (at least when the same doctor is following the same patient over time and when the same doctor is responsible for prescribing sick leave)” And finally, we have added more description of to whom findings apply, that no treatment options exist and that more research is needed, p. 17: “However, it should be noted that not everyone with a positive test will visit primary care with complaints, and treatment options are currently limited. We have previously reported that the prevalence of common medical complaints and health care visits following COVID-19 is elevated particularly 1-3 months after positive test.^{27,36} A small proportion of the individuals visiting primary care during 1-3 months post covid will still need care at 4-6 months post covid, however it is unclear what care would be helpful for this group of individuals. As such, the proposed checklist may be useful among individuals testing positive who are symptomatic, i.e., individuals visiting their doctor with complaints in the acute and/or sub-acute COVID-19 phase, when more treatment options are available. We believe this potential clinical usefulness of our findings as well as timely treatment options should be further investigated in future studies.”
4	- How well is long-Covid diagnosed by doctors and recorded in Norway? Were patients believed from early in the pandemic? How has prevalence/recording changed over time?	We agree the diagnostic practices deserve more attention in our work. Unfortunately, we have no overview over the reliability and validity of a long-covid diagnosis in primary care, which is partly due to	We have added the following to the Methods section, p. 5: “Still, we made use of a diagnostic coding practice that was introduced during the pandemic and therefore was relatively new to primary care physicians.

		long-covid being a condition characterized by different complaints in different combinations in different individuals, but also due to no studies yet being performed on the topic. However, it is reasonable to believe that long-covid was not expected by doctors and not recorded in Norway, at least for the first half year in 2020.	Indeed, the use of the codes as described above was limited in the beginning of the pandemic (when both the post-COVID condition was new, and also coding practices were new), before slowly rising and reaching its top in March 2022 (Supplementary Figure 1).” We have also performed a sensitivity analysis where we restrict our inclusion period to after December 2020 (Supplementary Figure 10), this is now referred to in the Statistical analyses section, (p. 7), as well as Results section (p. 11), and the discussion section (p. 18). In the results section, for example, we now write, p. 11: “In the supplementary material we show that our main results were robust across different sample selections: including hospitalized individuals (Supplementary Figure 7), including individuals with reinfection within 180 days (Supplementary Figure 8), including individuals either hospitalized and/or reinfected within 180 days (Supplementary Figure 9) and including individuals who were infected after the initial pandemic phase (Supplementary Figure 10).”
5	Patient groups prefer self-diagnosis as a definition.	We agree.	We have added the following to the discussion section, p. 18: “Moreover, patient groups prefer self-diagnosis as a definition, and we believe our findings need to be replicated and/or nuanced in future studies using patient-reported outcome measures.”
6	- According to the flow chart 8.9% of the population aged 30-70 were infected with Covid-19 in a two year period early in the pandemic. Even if you allow for repeat infections this does not tally with WHO figures for Norway https://covid19.who.int/region/euro/country/no	Thank you for pointing out this divergence. Our infections were retrieved directly from the official infection register, which also was the basis for the numbers reported to the WHO. A comment on the divergence: According the WHO graphic, there were 767,052 confirmed cases in total by January 23rd 2022. The study inclusion started July 1st 2020, and 8925 cases by June 29th. This (767,052-8925) is a considerably larger number than reported in our paper (238,001 individuals). However, there are good reasons for this divergence. First, by restricting our sample to ages 30-70, and living in the country at the start of 2017, the base population was reduced from ~5.6 million to ~2.67 million. Second, there was significant age differences in the likelihood of infection during different phases of the pandemic. Specifically, younger age groups were overrepresented among the	The reasons for divergence can be inferred from the flowchart, i.e., no revision was performed.

		infected during the Omicron wave, which also had the largest surge in infections. ¹	
7	Minor issues/points of clarification - Why was the age range limited to 30-70 years?	We agree this could be clarified. The age range was limited to above 30 years as by this age, most individuals will have reached their highest education level and be in their working age, hence allowing for using education and income in our models in a consistent way. As regards the upper age limit at 70, the reason was two-fold. First, individuals above this age may be more likely to have ended their work career, making income measures less reliable. Second, above age 70 a substantial part of the population receive care in nursing homes, which are not reported to the included/available registers.	We have revised the following in the Methods section, p. 3 (changes in italics): “Our study population included all Norwegian residents aged between 30 and 70 years old (i.e., working age individuals) on Jan 1st, 2020, and who had their first positive SARS-CoV-2 PCR test, as registered in the Norwegian Surveillance System of Communicable Diseases, between July 1st 2020 and January 23rd 2022.”
8	- Was follow-up limited to 180 days?	Yes, the follow-up period was limited to 180 days and we agree this could be clarified.	We have added to the Methods section, p. 3: “Using a prospective cohort study design following individuals for up to 180 days after the first positive test, we utilized data from the Norwegian Emergency Preparedness Register, Beredt C19 (BC19).”
9	- Individuals who were hospitalised were excluded. Therefore, this is a study of mild infection. This should be made clear in the title, abstract, and conclusions.	We agree.	We have revised the title into: “Predictors of the post-COVID condition following mild SARS-CoV-2 infection” We have also revised the abstract into: “Whereas the nature of the post-COVID condition following mild acute COVID-19 is increasingly well described in the literature, knowledge of its risk factors, and whether it can be predicted, remains limited.” “To assess the predictability of post-COVID after mild initial disease, we use modern machine learning methods and find that pre-infection characteristics, combined with information on the SARS-CoV-2 virus type and vaccine status, to a considerable extent (AUC = 0.79, 95% CI 0.75-0.81) could predict the occurrence of post-COVID complaints in our sample.” Our revised conclusion reads: “Individuals with mild initial COVID-19 with a prior history of psychological, respiratory, or unspecified/general health problems, had a higher risk of developing post-COVID complaints. There was also an increased risk among women and those infected by the Wuhan-virus. When validated in other samples and settings, these findings may be used by clinicians and care

¹ Report by the Norwegian Institute of Public Health: <https://www.fhi.no/contentassets/8a971e7b0a3c4a06bdf381ab52e6157/vedlegg/2.-alle-ukerapporter-2022/ukerapport-uke-8-21.02---27.02.22.pdf>

			providers to inform about the prognosis after COVID-19 regarding the development of the post-covid condition.”
10	- Table 1 only lists some symptoms of long-Covid. If these are the only ones under consideration it should be listed as a limitation. How about others, for example headache?	We agree this should be listed as a potential limitation.	We have added to the discussion section, p.18: “Moreover, patient groups prefer self-diagnosis as a definition, and we believe our findings need to be replicated and/or nuanced in future studies using patient-reported outcome measures. Along this line, there may be important post-covid complaints not studied here. For example, loss of taste and smell are commonly reported among patients ^{1,37} but could not be studied here because of low numbers.”
11	- Page 4 “Medical recording to the National registries is mandated by law in Norway, ensuring no missing data in our study”. No missingness cannot be guaranteed. There will be individual variation in diagnostic practice, which should be mentioned in the discussion.	We agree with the reviewer.	We have omitted the claim about missingness in the methods section, and the sentence now reads: “Medical recording to the National registries is mandated by law in Norway, reducing potential bias due to missing data in our study.” We have also added to the discussion section as a potential limitation, p. 18: “Still, there may be individual variations in coding practices, which might have influenced results.”
12	- Are there any data on ethnicity? A binary immigrant/non-immigrant variable is crude. Immigrants will be a heterogeneous group.	We agree this question might be of importance, however we have no available data on ethnicity and a thorough study of ethnicity is beyond the scope of our work. We think that classifying individuals on basis of ethnicity in research is complex and can be controversial - Please see discussion on the topic at the National Research Ethics Committee https://www.forskningsetikk.no/en/resources/the-research-ethics-library/research-on-particular-groups/ethnic-groups/	No further action taken.
13	- Is income recorded at the individual or household level? Both measures will have a level of error.	We agree that important detail was lacking on income in the overview of predictors. We used total individual income as our measure. We have now described this explicitly in Table 2. For now we chose to not mention this as a limitation since the individual income measure is precise, and any potential “error” would be related to what underlying dimension it fails to capture. We are certainly willing to reconsider this decision in case you disagree.	We have added the following to Table 2, overview of predictors: “Birth cohort- and gender-stratified income quintile, i.e., 5 categories based on the individual annual income.”
14	- Is number of primary care consultations a binary variable, or are there more categories? There is a big difference between consulting primary care	We agree this should be clarified. We chose to use a binary variable for whether or not the individual had visited their physician for each type of health problem,	We have revised Table 2 into:

	once over a period of a few years, versus regularly.	represented by ICPC2 chapters during the years 2017-19. Our data allowed for counting the number of visits, however we chose a binary coding to allow for a more transparent interpretation of the odds ratios in the predictive models.	“For each chapter in the International Classification of Primary Care (ICPC-2) coding system we created a categorical variable indicating whether the individual had one or more registered consultations in the period 2017-2019.”
15	- Please add reference groups to all odds ratio plots. It does not make sense to have separate odds ratios for (for example) males and females. One should be the referent unless I am misunderstanding the analysis.	We agree there is a need to clarify what were the reference group for each reported odds ratio. To enable the inclusion of all predictors in one figure, with a similar interpretation of the vertical axis (reference group), each predictor was included in the model as a binary variable, even if it made part of a multivariate variable taking multiple levels. The reported ORs were based on binary variables, and we used “everyone else” as the reference group in all analyses. However, in multivariate models using mutually exclusive groups as controls, such as age groups, income groups, and educational groups, in the models studying healthcare utilization, one group-level had to be removed for the model to estimate. In our case: age 60-70, 1st income quintile, and primary education. However, the reported OR estimates relating to healthcare utilization are invariant to this choice of reference, as this is just a technicality of estimating the model. We have therefore chosen to mention this as it could cause the reader to become confused. We are certainly willing to change this, if the reviewer still thinks this should be mentioned explicitly.	We have added more explanation to the methods section, p. 7: “For a more standardized interpretation of predictor-specific incidence and odds ratios, we used “everyone else” as the reference group in all analyses. Thus, all predictors were added to the model as a binary 0/1 variable, where 1 represented having the characteristic of interest (for example Age group (50,60]) taking value 1), and 0 represented everyone else, not having the characteristic of interest (in the example, all other age groups, i.e. age groups [30,40], [40,50], [60,70] taking value 0). Likewise, for predictor Female, coded as 1, everyone else, who were typically categorized as Male, were coded as 0. As such, the odds ratio for females will be the inverse of the odds ratio for males and vice versa.” And to figure legends: “The figures show estimated ORs the post-COVID condition for each predictor when included as a binary variable into the model. The reference group (i.e. dashed vertical line, OR=1) for all predictors was “everyone else”, i.e. everyone not having the predictor or characteristic of interest.”
16	- Typo on page 2 “16% higher risk among those most socioeconomically deprived” should be 11%.	Thank you for noticing.	Corrected, on p. 2.
17	- Typo on page 9 “had a psychological diagnosis were 121 percent more likely” should be 12%.	The OR was 2.12, we updated to “approximately twice as likely” since the low base rate in practice allows for a relative risk interpretation.	Corrected to “approximately twice as likely”, p. 11.
#	Reviewer 3	Our response	Action
1	Thank you for asking me to review this paper. The authors use data from a national Norwegian registry, identifying people who tested PCR positive for COVID-19 between July 2020 and Jan 2022 and analysing socioeconomic, demographic, vaccination and healthcare-utilisation data to look for predictors of post-COVID condition. They identify several factors that have been well established in other studies		

	(female sex, vaccination status, COVID variant) and the authors focus on the findings relating to comorbidities, which show elevated risk for a number of conditions (as identified by pre-covid healthcare use).		
2	I have some reservations about the findings and these are mostly related to the definition of the outcome. The binary post-covid condition outcome is defined as a ICPC-2 code of R992 (confirmed COVID-19) plus any one of a range of documented conditions (fatigue, cough, palpitations, shortness of breath, anxiety, depression, brain fog, musculoskeletal pain). What is unclear is how many of these conditions were pre-existing, and the extent to which these conditions can be confidently linked to the COVID-19 infection. Clearly it is not a requirement for the patient not to have experienced the condition before, as pre-existing anxiety and depression are identified as risk factors. Is it down to the physician's judgement as to whether the condition has been exacerbated by COVID? Whatever the answer, this needs some clarification and ideally some more interrogation. Otherwise it might be argued that you're concluding that eg having depression before COVID is associated with having depression after COVID.	Thank you for this important comment. We agree that more explanation is required.	Please see specific actions taken beneath, to each of the suggestions.
3	My suggestions would be: > Include a paragraph explaining in more detail what the criteria were for the physicians to associate the health problems with COVID-19 infection.	We agree with the reviewer.	We have added to the methods section, on p. 4: "The main outcome of interest was having the post-COVID condition (yes/no) as recorded by a general practitioner (GP) in primary or emergency care by the International Classification of Primary Care (ICPC-2) code. From May 4th 2020, primary care physicians were instructed to use the code R992 diagnosis for patients with COVID-19 disease. Persistent complaints after COVID-19 were coded by an R992 code together with at least one code for a persistent symptom, for example fatigue or pain. ²⁴ For example, if a patient reported to be struggling with fatigue after the infection, it was coded with R992 together with A04 (weakness/tiredness). Correspondingly, if the complaint was shortness of breath, it was coded with R992 and R02. This coding for persistent complaints was possible for primary care physicians to use at any time during the pandemic. However, an official recommendation to do so was provided by national health authorities from April 1st 2021. The recommendation stated that persistent COVID-19 complaints should be coded by primary care physicians based on patient history of persistent complaints and an earlier, confirmed COVID-19. In our study, we assessed physician-reported post-COVID condition for one or more of several long-term

			symptoms after a SARS-CoV2 infection as described in Table 1 25, if they occurred in the time range 90 to 180 days after the first positive test. As such, our definition is in accordance with the World Health Organization’s definition of the post-covid conditions (covid-like complaints present 3 months after infection).⁸⁴ We also provide a timeline overview of the use of R992 + complaint, in the supplementary file (Supplementary Figure 1), and have added the following to the description of the outcome in the methods section (p. 5): “Still, we made use of a diagnostic coding practice that was introduced during the pandemic and therefore was relatively new to primary care physicians. Indeed, the use of the codes as described above was limited in the beginning of the pandemic (when both the post-COVID condition was new, and also coding practices were new), before slowly rising and reaching its top in March 2022 (Supplementary Figure 1).” Lastly, in strengths and weaknesses, we have added/revised, with reference to a robustness check, p. 18: “Lastly, the post-COVID condition was a new phenomenon in the early phases of the pandemic and general practitioners may not have known how to interpret, or code, the symptoms reported by their patients. Although it was possible to register an R992 code together with a persistent complaint, the primary care physicians might not have done so. The operationalization chosen in this study is in line with the official guide given to general practitioners in April 2021 and in accordance with the WHO definition of the post-COVID condition. We found similar results in our sensitivity analysis where inclusion was started in January 2021, with corresponding potential post-COVID cases from April 2021 (Supplementary Figure 10). Still, there may be individual variations in coding practices, which might have influenced results.”
4	> Include a table showing a breakdown of coded post-covid health problems among the 908 people with post-covid condition. Ideally, also show how many of these people had these conditions before their COVID-19 diagnosis.	We agree such a table is useful.	Please see Supplementary Table 1, which is now referred to in the results section, p. 8-9: “In total, 0.42% (N = 908) were classified as having a post-COVID condition (main outcome). Among these participants, 206 (21%) were classified as experiencing post-COVID respiratory complaints, while 584 (60%) were classified as experiencing post-COVID fatigue (cf. Supplementary Table 1). Of the 206 individuals with post-COVID respiratory complaints, 191 (93%) were new onsets compared to the period between 2017 and 2019 (Supplementary Table 1). Similarly, out of the 584 individuals with fatigue, 444 (76%) had new onsets of fatigue complaints. Hence, for most participants these complaints were more likely to be due to the infection and not already preexisting conditions.”
5	> A useful sensitivity analysis would be to remove, for example, people with anxiety and	We agree this sensitivity analysis is useful.	We have run the suggested analyses.

	depression related post-covid condition and see if pre-existing anxiety and depression are still identified as risk factors.		The sensitivity analyses have now been specified in the methods section, p. 4: “As a robustness check, because individuals with anxiety and/or depression might be more prone than others to seek medical care due to health concerns also for physical health issues²⁸, we also examined how the results were affected when recoding individuals with anxiety and depression post-COVID symptoms as non-post-COVID cases.” Accordingly, we have added a description of findings in our results section, p. 11: “Approximately similar estimates were found in analyses of our secondary outcome measures (post-COVID respiratory complaints and post-COVID fatigue; Supplementary Figures 4 and 5). In additional robustness analysis (Supplementary Figure 6), we recoded individuals with post-COVID anxiety and depression as non-post-COVID cases, the OR for Psychological health problems was then 1.78 (95% CI 1.53-2.08).”
6	The methods describe logistic regression models and odds ratios but the figure legends refer to Cox models.	Thank you for noticing and making us aware of this error. Logistic regression is correct.	Corrected.
7	The phrasing of the specification of the adjusted logistic regression models was a little confusing. “When studying demographic, sociodemographic and vaccine status information, the multivariate models adjusted for healthcare utilization prior to infection” – does this mean that they *only* adjust on healthcare utilization? Or that they adjust on healthcare utilization *and* sociodemographic/vaccine variables?	We agree this was not a sufficiently clear phrasing of the model specification.	In the statistical analysis section we now write, p. 6-7: “When studying the risk related to demographic and sociodemographic characteristics, and vaccine status, we ran separate models for each characteristic while adjusting for the healthcare utilization prior to infection (2017-2019). To illustrate, the adjusted model for a specific age group shows the risk adjusted for health care utilization history.”
8	How were the signs on the RF variable importance determined?	The sign of the variables in the random forest models were determined by switching on/off each predictor and checking whether it increased/decreased the average likelihood of the outcome.	We have added to the note in Figure 2, p. 14: “The signs of the predictors in the Random Forest were determined by comparing average sample likelihoods when recoding the predictor in question on/off for all individuals. If the average sample likelihood increased, the sign was coded as “POS”, otherwise “NEG”.”
9	In the bivariate analyses, what were the reference categories? For instance, in the income categories, are you comparing 80-100 percentile vs pooled 0-80 percentiles? If so, I think it would be more interpretable to regress all percentile categories together as a single categorical variable and choose a reference category.	Thank you, we agree the reference category needs a more thorough explanation.	We have added to the methods section, p. 7: “For a more standardized interpretation of predictor-specific incidence and odds ratios, we used “everyone else” as the reference group in all analyses. Thus, all predictors were added to the model as a binary 0/1 variable, where 1 represented having the characteristic of interest (for example Age group (50,60]) taking value 1), and 0 represented everyone else, not having the characteristic of interest (in the example, all other age groups, i.e. age groups [30,40], [40,50], [60,70] taking value 0). Likewise, for predictor Female, coded as 1, everyone else, who were typically categorized as Male, were coded as 0. As such, the odds ratio for females will be the inverse of the odds ratio for males and vice versa.”

			And to the figure legends: “The figures show estimated ORs the post-COVID condition for each predictor when included as a binary variable into the model. The reference group (i.e. dashed vertical line, OR=1) for all predictors was “everyone else”, i.e .everyone not having the predictor or characteristic of interest.”
10	Relating to the above, in the discussion you say that you observed no clear socioeconomic gradient. However, it does seem like there is a gradient, especially in income. The effect might be more apparent in a model with a categorical predictor.	Thank you, we agree.	We have revised and nuanced our discussion section, p. 15: “We found indications of a U-shaped association between income and the post-COVID condition, i.e., that individuals with middle income (40th to 80th percentile) has higher odds for having the post-COVID condition than other individuals. Moreover, individuals with low university education had a higher odds of the post-COVID condition when compared to individuals with other education levels. These findings of socioeconomic gradient are contradictory to findings reported in other studies.^{6,22} It should be noted, however, that the absolute differences in resources between the top and bottom of socioeconomic distributions differ significantly between countries, making cross-country comparisons difficult to interpret.”

REVIEWERS' COMMENTS

Reviewer #1 (Remarks to the Author):

My questions and concerns have been well addressed. Suggested revisions have also been made appropriately. I have no further comments. Good work. Thanks.

Reviewer #2 (Remarks to the Author):

Thank you for addressing my comments and amending the manuscript where appropriate. I am happy with the revised version and have nothing to add.

Reviewer #3 (Remarks to the Author):

The authors have made significant efforts to address my comments and the comments from other reviewers. This includes running sensitivity analyses that I think reinforce the findings, and deal with my primary concern about the first version (that there was some 'X in the Y' – ie pre-existing conditions were potentially being included in the post-covid condition definition).

I have no further comments – thank you for inviting me to review this interesting work.